# FROM LEADS TO LATENTS: ATTENTION-DRIVEN MASKED AUTOENCODER FOR ECG TIMES SERIES

## ABSTRACT

Electrocardiograms (ECGs) are among the most widely available clinical signals and play a central role in cardiovascular diagnosis. While recent foundation models have shown promise for learning transferable ECG representations, most existing pretraining approaches treat leads as independent channels and fail to explicitly leverage their strong structural redundancy. We introduce the latent attention masked autoencoder (LAMAE) framework that directly exploits this structure by learning cross-lead connection mechanisms during self-supervised pretraining. Our approach models higher-order interactions across leads through latent attention, enabling permutation-invariant aggregation and adaptive weighting of lead-specific representations. We provide empirical evidence on the Mimic-IV-ECG database that leveraging the cross-lead connection constitutes an effective form of structural supervision, improving representation quality and transferability. Our method shows strong performance in predicting ICD-10 codes, outperforming independent-lead masked modeling and alignment-based baselines.

## 1 INTRODUCTION

Cardiovascular diseases remain among the leading causes of death worldwide (Roth et al., 2025). Clinical diagnosis and monitoring increasingly rely on multimodal data streams including imaging, clinical notes, lab tests, and physiological signals, among which the electrocardiogram (ECG) is the most ubiquitous modality due to its cost-efficiency, non-invasiveness, and mature clinical interpretation pipelines (Kashou et al., 2023). Its automated diagnosis has been dominated for decades by expert-crafted features coupled with classical classifiers (Liu et al., 2014; Chen et al., 2018). Over the last years, deep learning has largely shifted the field toward end-to-end learning from raw ECGs, with convolutional neural networks (CNNs) (Ribeiro et al., 2020) and recurrent models (Übeyli, 2010) as the predominant architectures for many clinical tasks (Sau et al., 2024; Hannun et al., 2019).

More recently, foundation models have emerged as a compelling direction to reduce reliance on expensive medical labels and enable transfer across tasks and cohorts (Moor et al., 2023; Tian et al., 2024). Yet, frontier general-purpose models still lag behind domain experts on clinical benchmarks and remain costly to adapt or deploy in practice (Khan et al., 2025). A key reason is that most pretraining pipelines remain largely oblivious to domain structure, particularly in ECG time series, where such structure is particularly explicit. This structure is not a nuisance; it is an intrinsic self-supervisory signal that modern pretraining objectives rarely exploit directly and that motivates cross-lead representations rather than independent-lead designs.

Self-supervised learning (SSL) offers a scalable alternative to label-heavy supervision in medicine (Azizi et al., 2021; Moody et al., 2025; Manduchi et al., 2023). Masked autoencoders (MAEs) (He et al., 2022) have gained momentum by reconstructing missing content from sparse context, encouraging learning robust, transferable representations. In ECG specifically, masked modeling has been explored on an independent-lead basis (Na et al., 2024) and with language-inspired tokenization schemes (Jin et al., 2024). Yet, existing methods often tokenize using lead-specific encoders or treat leads as quasi-independent "channels", limiting their ability to learn cross-lead correspondence.

Leveraging the coherence of medical datasets, clinical recordings often come as structured multi-view observations that share anatomy and semantics across views. Exploiting this structure via multiview contrast, cross-modal alignment, or multitask learning (Laguna et al., 2025) can improve robustness and label efficiency in multimodal models (Mo & Liang, 2024; Pellegrini et al., 2025;

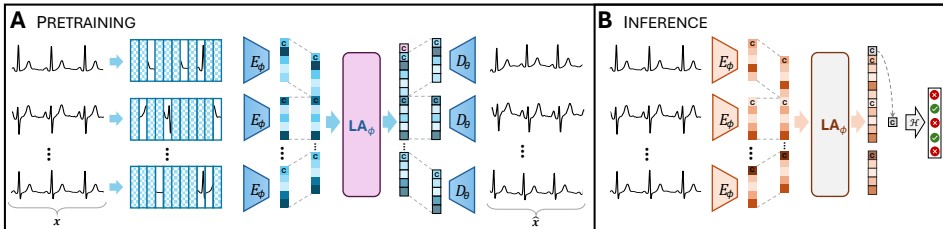

Figure 1: **Framework overview.** (left) Each ECG is separated into 12 leads, which are encoded separately and subsequently processed jointly through a latent attention transformer. The training objective is masked reconstruction. (right) The predictions are based on the latent attention's CLS token.

Chen et al., 2024). Recently, Erlacher et al. (2025) combined multi-lead MAE reconstruction with a lead-alignment objective to enforce cross-lead consistency. While effective, such a pairwise alignment comes with limitations (Tschannen et al., 2023), which complicates design and may under-utilize richer, higher-order relationships among leads. In contrast, attention-based aggregation is a natural fit for structured latent sets: it supports permutation-invariant processing of variable-size collections while learning which elements are most informative (Lee et al., 2019) and has been shown to provide interpretable, instance-weighted summaries in related weakly supervised settings (Ilse et al., 2018).

In this work, we propose a multi-lead MAE framework that directly capitalizes on ECG structure by learning cross-lead connection mechanisms and modeling higher-order lead interactions by integrating latent attention. Concretely, our contribution is four-fold: (i) we introduce a multi-lead MAE with explicit cross-lead connection learning that leverages intrinsic redundancy across leads; (ii) we enhance the framework with latent attention to capture higher-order dependencies beyond pairwise alignment; (iii) we provide empirical arguments for cross-lead connection learning as a scalable form of structural supervision; and (iv) we demonstrate broad clinical translation spanning coarse ICD-based phenotyping to fine-grained disease classification.

## 2 METHODS

We assume an ECG dataset $\mathbb{X} = \{\boldsymbol{X}^{(i)}\}_{i=1}^N$, where $N$ is the number of ECG recordings in the dataset, $\boldsymbol{X}^{(i)} = \{\boldsymbol{x}_l^{(i)}\}_{l \in \mathbb{L}}$, $\mathbb{L}$ is the set of leads (e.g. 12 leads in our dataset $\mathbb{L} = \{\ell_I, \ell_{II}, \ell_{III}, \ell_{a_{VR}}, \ell_{a_{VL}}, \ell_{a_{VF}}, \ell_{V_1}, \ell_{V_2}, \ell_{V_3}, \ell_{V_4}, \ell_{V_5}, \ell_{V_6}\}$ for our dataset (Strodthoff et al., 2024a) or see section A.

The proposed work extends the Masked Autoencoder method (He et al., 2022) to the ECG domain by introducing a self-attention module in the latent space, i.e., between the encoders $E_\phi$ and the decoders $D_\theta$. We call the new module *Latent Attention* (LA).

### 2.1 LATENT ATTENTION FOR MULTI-LEAD INTEGRATION

Our latent attention module, inspired by Ilse et al. (2018); Lee et al. (2019), learns the correlation and shared information between different leads but is flexible enough to not having to merge information between leads in case it would be supoptimal.

We design the latent attention module as a multi-head, multi-layer self-attention block using an additional CLS token similar to the ViT architecture (Kolesnikov et al., 2021).

$$\boldsymbol{Z}_{out}^{(i)} = LA_\phi(\boldsymbol{Z}_{\text{vis}}^{(i)}) = LA_\phi(M_{\text{LA}}(\boldsymbol{Z}^{(i)})) = LA_\phi(M_{\text{LA}}(\{\boldsymbol{z}_l^{(i)}\}_{l \in \mathbb{L}})) \tag{1}$$

Similar to encoder $E_\phi$ in the MAE, we apply a random mask $M_{\text{LA}}(\cdot)$ to the input embeddings $\boldsymbol{Z}^{(i)}$, i.e., $\boldsymbol{Z}_{\text{vis}}^{(i)} = M_{\text{LA}}(\boldsymbol{Z}^{(i)})$ with a masking ratio $\alpha_{\text{LA}}$. See section 2.2 for details on the masking process.

### 2.2 ECG-LAMAE

Different to previous works (Na et al., 2024; Jin et al., 2024), we use per-lead encoders $E_\phi$ and decoders $D_\theta$ with shared weights $\phi$ and $\theta$. The latent attention module allows the model to learn the connection between the different leads to extract more meaningful information.

Table 1: ICD-10 code prediction performance by hierarchical group under linear probing of the corresponding backbones from the studied models. Best results in **bold** and second best in *italics*.

| | **Ours** | | **Baselines** | | | |
|---|---|---|---|---|---|---|
| **ICD hierarchy** | **LAMAE** | **LAMAE$_E$** | **Scratch** | **MMVM** | **Ind$_P$** | **Ind$_S$** |
| **IX** | *0.8345* | 0.8340 | 0.7715 | 0.6771 | **0.8380** | 0.7776 |
| IX.I05–I09 | **0.8317** | *0.8278* | 0.7408 | 0.5754 | 0.8259 | 0.7564 |
| I07 | *0.8772* | **0.8802** | 0.8091 | 0.6804 | 0.8669 | 0.8310 |
| I08 | **0.8272** | *0.8263* | 0.7523 | 0.6755 | 0.8235 | 0.7676 |
| IX.I10–I1A | *0.7549* | **0.7570** | 0.7043 | 0.5861 | 0.7506 | 0.7092 |
| I11 | 0.8103 | *0.8124* | 0.7491 | 0.6283 | **0.8163** | 0.7602 |
| I13 | *0.8651* | **0.8704** | 0.8106 | 0.6826 | 0.8655 | 0.8194 |
| IX.I20–I25 | **0.8046** | *0.7995* | 0.7425 | 0.6182 | 0.7939 | 0.7508 |
| I20 | 0.7880 | **0.7963** | 0.7147 | 0.6574 | *0.7934* | 0.7324 |
| I21 | 0.8229 | **0.8402** | 0.7465 | 0.5585 | *0.8265* | 0.7542 |
| IX.I26–I28 | 0.7399 | *0.7474* | 0.6948 | 0.5785 | **0.7499** | 0.6977 |
| IX.I30–I5A | 0.8624 | 0.8624 | 0.7955 | 0.6321 | **0.8665** | 0.8010 |
| I35 | 0.7981 | **0.8050** | 0.7365 | 0.6246 | *0.7983* | 0.7423 |
| I42 | *0.8772* | **0.8838** | 0.8396 | 0.7193 | 0.8731 | 0.8458 |
| I44 | 0.9019 | *0.9050* | 0.8439 | 0.7591 | **0.9091** | 0.8467 |
| I45 | *0.8407* | **0.8493** | 0.7766 | 0.6646 | 0.8392 | 0.7828 |
| I46 | 0.8336 | *0.8481* | 0.7697 | 0.6562 | **0.8523** | 0.7809 |
| I47 | *0.8051* | 0.8023 | 0.7409 | 0.6480 | **0.8140** | 0.7461 |
| I48 | 0.8837 | *0.8861* | 0.7852 | 0.6952 | **0.8927** | 0.7933 |
| I50 | *0.8720* | **0.8747** | 0.8166 | 0.6169 | *0.8720* | 0.8235 |
| IX.I60–I69 | *0.6694* | *0.6723* | 0.6290 | 0.5473 | **0.6823** | 0.6348 |
| IX.I70–I79 | *0.7416* | **0.7441** | 0.6942 | 0.5956 | 0.7440 | 0.6974 |
| IX.I80–I89 | 0.6890 | *0.6949* | 0.6349 | 0.5731 | **0.6985** | 0.6324 |
| IX.I95–I99 | 0.6772 | *0.6782* | 0.6224 | 0.5519 | **0.6943** | 0.6275 |

As in the standard MAE implementation, we only feed the visible tokens $\mathcal{T}_{\text{vis}}$ to the encoders $E_\phi$, where $\mathcal{T}_{\text{vis}} \subseteq \{1, \ldots, T\}$ are the indices of the visible patches after applying $M_{\text{E}}(\cdot)$, i.e., $\boldsymbol{x}_{l_{\text{vis}}}^{(i)} = M_{\text{E}}(\boldsymbol{x}_l^{(i)})$. We therefore have $T_{\text{vis}} = |\mathcal{T}_{\text{vis}}| = (1 - \alpha_{\text{E}}) \cdot T$, where $T$ is the total number of input tokens.

Using our latent attention module, we have the following objective function

$$\mathcal{L}\left(\boldsymbol{X}^{(i)}\right) = \frac{1}{\alpha_{\text{E}}} \frac{1}{|\mathbb{L}|} \sum_{l \in \mathbb{L}} \sum_{t \notin \mathcal{T}_{\text{vis}}} \left\| \boldsymbol{x}_{l_t}^{(i)} - \hat{\boldsymbol{x}}_{l_t}^{(i)} \right\|_2^2, \quad \text{where} \quad \hat{\boldsymbol{x}}_l^{(i)} = D_\phi(LA(\boldsymbol{Z}_{\text{vis}}^{(i)})_l)$$

and $\boldsymbol{Z}_{\text{vis}}^{(i)} = M_{\text{LA}}(\{\boldsymbol{z}_l^{(i)}\}_{l \in \mathbb{L}})$ is the set of all non-masked latent tokens $\boldsymbol{z}_l^{(i)}$ coming from all leads $\boldsymbol{x}_l^{(i)}$, i.e., $\boldsymbol{z}_l^{(i)} = E_\phi(\boldsymbol{x}_l^{(i)})$. This architecture supports the extraction of relevant information of each lead, combined with subsequent merging of the information in the latent attention module for a global representation captured within the CLS token. This token is then used for downstream tasks, as depicted in fig. 1 (right).

## 3 EXPERIMENTS AND RESULTS

We evaluated multi-label ICD-10 prediction from 12-lead ECGs across Chapter IX (I00–I99), spanning valvular disease, hypertensive disease, ischemic syndromes and myocardial infarction, pulmonary circulation disorders, cardiomyopathies, conduction disease, atrial fibrillation/flutter, heart failure, and vascular/cerebrovascular conditions (table 1 and appendix sections A to C). Overall, LAMAE-based models achieve strong performance across granularities, with chapter-level AUROC $\approx 0.85$ (after fine-tuning; table 4) and competitive linear-probing results (IX: 0.834; table 3). Performance is highest for ECG-salient phenotypes, notably conduction and rhythm disorders (e.g., I44 fine-tuning up to 0.9097; I48 up to 0.9016) and acute myocardial infarction subtypes (I21.* often >0.93; I210 up to 0.9749), consistent with stereotyped waveform signatures (PR/QRS abnormalities, irregular rhythm, ST/T changes). In contrast, broader vascular and cerebrovascular groupings (I60–I69, I80–I89, I95–I99) are harder from waveform-only inputs (fine-tuning ~0.69–0.72), plausibly reflecting weaker direct ECG imprint and higher label/context heterogeneity. Table 1 is supplemented by an exploration of fine-tuning performance (Tab. 4), which details the greater improvements over linear probing and highlights the benefit of the pretrained backbone for downstream adaptationm, as well as the fine-grained hierarchy results for linear probing in Tab.3.

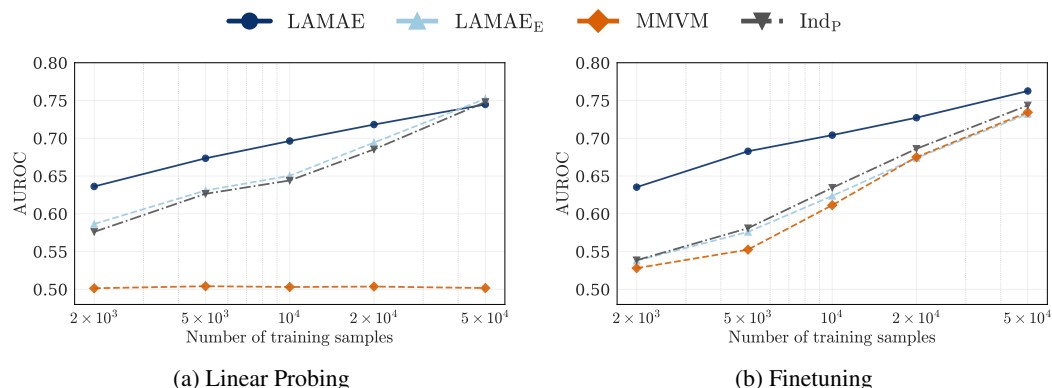

(a) Linear Probing  (b) Finetuning

Figure 2: **Label efficiency under finetuning.** Performance curves of the macro-averaged AUROC over all 228 Chapter IX codes, as a function of the number of training studies used for finetuning.

Scaling experiments show that gains from structure-informed pretraining concentrate in scarce-data regimes (Fig. 2). Under linear probing and full fine-tuning, LAMAE outperforms scratch-trained and simpler baselines most strongly at small pretraining set sizes, with gaps narrowing only in large regimes (e.g., $\gtrsim$50k samples; Fig. 2). This supports latent attention as a structure-aware fusion mechanism over correlated lead projections: it can exploit redundancy to encode shared physiology, yielding more sample-efficient representations when curated cardiology datasets are limited.

A broader ICD-10 benchmarking study is provided by Strodthoff et al. (2024b). While not directly comparable, our fine-tuned AUROCs are in a similar range or higher for several overlapping, ECG-identifiable codes, including IX (0.8495), I132 (0.9119), I210 (0.9632), I447 (0.9452), and AF-related subcodes such as I481 (0.8902) and I482 (0.9312) (table 3). Strodthoff et al. (2024b) likewise reports strong results for conduction/AF-related codes (e.g., I440 and AF groupings). For the global burden of atrial fibrillation (ICD48 and 48.*; (Chugh et al., 2014)), our performance is superior to prior task-specific studies that report AUROCs around 0.82–0.85 across external cohorts with CNN-based models (Brant et al., 2025), and 0.67–0.8 using demographics or NN-extracted features on a 1-day ECG recording (Gadaleta et al., 2023). This is broadly consistent with AF being learnable yet sensitive to cohort shift and label timing. For conduction/heart block phenotypes related to I44 and sub-groups, reported performance varies widely across clinical settings ranging 0.594 to 0.889 (Sau et al., 2025), and our results with AUROC above 0.9 suggest that multi-lead structural pretraining can yield robust discrimination even under limited downstream data.

Limitations include the imperfect nature of ICD labels as proxies for physiology and the restricted clinical context available to waveform-only models, particularly for vascular/cerebrovascular diagnoses. Nevertheless, the consistent low-data gains and strong performance on ECG-salient phenotypes indicate that explicitly leveraging cross-lead structure via latent attention is a practical route toward more transferable ECG foundation representations.

## 4 CONCLUSION

We introduced LAMAE, a multi-lead masked autoencoder that injects structure into ECG pretraining via latent attention over lead-specific latents. Across a broad Chapter IX ICD-10 hierarchy, LAMAE yields strong AUROC under both linear probing and fine-tuning, with the largest advantages in low-data regimes and in diagnoses where multi-lead interactions are central. These results support that exploiting cross-lead redundancy as structural supervision can improve sample efficiency and down-stream transfer, offering a scalable template for time-series foundation models, potentially beyond ECG, where observations naturally come as correlated sets of views. Even more, latent attention could serve as a general template wherever there is structure between measurement to be leveraged.

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

## A    MATERIALS

We conducted experiments on the MIMIC-IV-ECG-Ext-ICD resource (Strodthoff et al., 2024a), a PhysioNet (Goldberger et al., 2000)release that links raw 12-lead ECG waveforms from MIMIC-IV-ECG (Gow et al., 2023) to clinically grounded diagnostic labels from the corresponding MIMIC-IV Johnson et al. (2023) emergency department and inpatient records. Concretely, ECG acquisition timestamps are aligned with ED stays and hospital admissions to associate each recording with discharge diagnosis codes, providing ICD-10-CM label sets derived from routine clinical documentation rather than retrospective re-annotation. The dataset includes identifiers to retrieve additional clinical context (e.g., ED stay and hospital admission IDs), basic demographics (e.g., age-at-recording, sex), and fold assignments designed to avoid patient overlap for benchmarking and comparability across studies (Strodthoff et al., 2024a). In our study, only ECG raw waveforms and their paired ICD-10 code were used.

Following the benchmark framing introduced by Strodthoff et al. (2024b), we treat ICD-10-CM codes as multi-label targets at multiple granularities (chapter/block/category/subcategory), enabling evaluation from coarse phenotyping to fine-grained diagnosis. Where needed for consistency across label hierarchies, ICD codes may be normalized to a fixed digit format and expanded to include higher-level ancestors in the ICD tree, supporting hierarchical reporting and clinically meaningful aggregation (Strodthoff et al., 2024a;b).

## B    ON ICD CODES AND FURTHER DETAILS ON THOSE USED IN THIS STUDY

International Classification of Diseases (ICD) codes provide a standardized taxonomy for clinical diagnoses and are routinely used for billing, cohort definition, and large-scale observational research. In this work, we focus on ICD-10 Chapter IX (Diseases of the circulatory system; I00–I99), and report predictive performance at multiple levels of granularity: (i) the chapter-level aggregate, (ii) chapter blocks (e.g., I05–I09), (iii) 3-character categories (e.g., I07), and (iv) selected 4-character subcategories (e.g., I07.1; written as I071). This hierarchical evaluation reflects clinically meaningful groupings while enabling finer assessment of model behaviour on specific diagnoses. An extended description of the clinical meaning per code is included in table 2.

## C    SUPPLEMENTARY RESULTS: FINE-GRAINED ANALYSIS

**Fine-grained Classification Results.**    In table 3, we present an extended analysis of the fine-grained classification performance initially discussed in table 1 on linear probing. The results demonstrate that performance trends remain remarkably consistent across the ICD-10 hierarchy. Notably, the relative advantages of our proposed methods are preserved even as the classification task becomes more granular, confirming the robustness of the learned representations.

**Performance of LAMAE under Varying Supervision.**    Table 4 compares the performance of LAMAE and LAMAE$_E$ across both linear probing and fine-tuning regimes. While both models achieve competitive results under linear probing, fine-tuning LAMAE yields a significantly larger performance gain. This suggests that the pretrained backbone serves as a powerful initialization that can be further leveraged to maximize predictive accuracy when labeled data allows for full model updates.

Table 2: Clinical meaning of the ICD-10 Chapter IX codes reported in tables 1 and 3, shown with the same hierarchy.

| ICD hierarchy | Clinical description |
|---|---|
| **IX** | Diseases of the circulatory system (I00–I99). |
| IX.I05–I09 | Chronic rheumatic heart diseases (I05–I09). |
| I07 | Rheumatic tricuspid valve diseases. |
| I071 | I07.1 — Tricuspid (valve) insufficiency (rheumatic). |
| I078 | I07.8 — Other tricuspid valve diseases. |
| I08 | Multiple valve diseases (often rheumatic or unspecified origin). |
| I080 | I08.0 — Disorders of both mitral and aortic valves. |
| I081 | I08.1 — Disorders of both mitral and tricuspid valves. |
| I083 | I08.3 — Combined disorders of mitral, aortic and tricuspid valves. |
| IX.I10–I1A | Hypertensive diseases (I10–I15). |
| I11 | Hypertensive heart disease. |
| I13 | Hypertensive heart and renal disease. |
| I130 | I13.0 — Hypertensive heart and renal disease with (congestive) heart failure. |
| I132 | I13.2 — Hypertensive heart and renal disease with both (congestive) heart failure and renal failure. |
| IX.I20–I25 | Ischaemic heart diseases (I20–I25). |
| I20 | Angina pectoris. |
| I200 | I20.0 — Unstable angina. |
| I209 | I20.9 — Angina pectoris, unspecified. |
| I21 | Acute myocardial infarction. |
| I210 | I21.0 — Acute transmural myocardial infarction of anterior wall. |
| I211 | I21.1 — Acute transmural myocardial infarction of inferior wall. |
| I213 | I21.3 — Acute transmural myocardial infarction of unspecified site. |
| I214 | I21.4 — Acute subendocardial myocardial infarction. |
| IX.I26–I28 | Pulmonary heart disease and diseases of pulmonary circulation (I26–I28). |
| IX.I30–I5A | Other forms of heart disease (ICD block: I30–I52; reported here as I30–I5A). |
| I35 | Nonrheumatic aortic valve disorders. |
| I350 | I35.0 — Aortic (valve) stenosis. |
| I359 | I35.9 — Aortic valve disorder, unspecified. |
| I42 | Cardiomyopathy. |
| I420 | I42.0 — Dilated cardiomyopathy. |
| I428 | I42.8 — Other cardiomyopathies. |
| I429 | I42.9 — Cardiomyopathy, unspecified. |
| I44 | Atrioventricular and left bundle-branch block. |
| I440 | I44.0 — Atrioventricular block, first degree. |
| I441 | I44.1 — Atrioventricular block, second degree. |
| I442 | I44.2 — Atrioventricular block, complete. |
| I447 | I44.7 — Left bundle-branch block, unspecified. |
| I45 | Other conduction disorders. |
| I46 | Cardiac arrest. |
| I47 | Paroxysmal tachycardia. |
| I48 | Atrial fibrillation and flutter. |
| I480 | I48.0 — Paroxysmal atrial fibrillation. |
| I481 | I48.1 — Persistent atrial fibrillation. |
| I482 | I48.2 — Chronic atrial fibrillation. |
| I50 | Heart failure. |
| IX.I60–I69 | Cerebrovascular diseases (I60–I69). |
| IX.I70–I79 | Diseases of arteries, arterioles and capillaries (I70–I79). |
| IX.I80–I89 | Diseases of veins, lymphatic vessels and lymph nodes, not elsewhere classified (I80–I89). |
| IX.I95–I99 | Other and unspecified disorders of the circulatory system (I95–I99). |

Table 3: Extended ICD-10 code prediction performance by hierarchical group in the models studied in table 1. AUROC is reported for linear probing on each corresponding backbone.

| ICD hierarchy | Ours | | Baselines | | | |
| --- | --- | --- | --- | --- | --- | --- |
| | LAMAE | LAMAE$_E$ | Scratch | MVMAE | Ind$_P$ | Ind$_S$ |
| **IX** | *0.8345* | 0.8340 | 0.7715 | 0.6771 | **0.8380** | 0.7776 |
| IX.I05–I09 | **0.8317** | *0.8278* | 0.7408 | 0.5754 | 0.8259 | 0.7564 |
| I07 | *0.8772* | **0.8802** | 0.8091 | 0.6804 | 0.8669 | 0.8310 |
| I071 | **0.8830** | *0.8740* | 0.8236 | 0.6450 | 0.8609 | 0.8585 |
| I078 | *0.8818* | **0.8977** | 0.8110 | 0.5145 | 0.8896 | 0.8345 |
| I08 | **0.8272** | 0.8235 | 0.7523 | 0.6755 | *0.8263* | 0.7676 |
| I080 | *0.8345* | 0.8318 | 0.7418 | 0.6619 | **0.8390** | 0.7513 |
| I081 | **0.8704** | *0.8695* | 0.7719 | 0.7272 | 0.8556 | 0.7992 |
| I083 | **0.8852** | 0.8573 | 0.8421 | 0.7311 | *0.8847* | 0.8742 |
| IX.I10–I1A | *0.7549* | **0.7570** | 0.7043 | 0.5861 | 0.7506 | 0.7092 |
| I11 | 0.8103 | *0.8124* | 0.7491 | 0.6283 | **0.8163** | 0.7602 |
| I13 | *0.8651* | **0.8704** | 0.8106 | 0.6826 | 0.8655 | 0.8194 |
| I130 | *0.8629* | **0.8687** | 0.8183 | 0.7566 | 0.8668 | 0.8254 |
| I132 | *0.9009* | 0.9096 | 0.7878 | 0.6384 | **0.9171** | 0.8089 |
| IX.I20–I25 | *0.7995* | **0.8046** | 0.7425 | 0.6182 | 0.7939 | 0.7508 |
| I20 | 0.7880 | **0.7963** | 0.7147 | 0.6574 | *0.7934* | 0.7324 |
| I200 | 0.8236 | **0.8269** | 0.7366 | 0.6716 | *0.8267* | 0.7522 |
| I209 | *0.7925* | **0.8075** | 0.7224 | 0.6796 | 0.7870 | 0.7414 |
| I21 | 0.8229 | **0.8402** | 0.7465 | 0.5585 | *0.8265* | 0.7542 |
| I210 | *0.9330* | **0.9632** | 0.9197 | 0.6276 | 0.9343 | 0.9143 |
| I211 | *0.9370* | **0.9588** | 0.8049 | 0.6604 | 0.9269 | 0.8433 |
| I213 | 0.8649 | **0.8996** | 0.8097 | 0.6387 | *0.8814* | 0.8010 |
| I214 | 0.8043 | **0.8117** | 0.7350 | 0.6018 | *0.8063* | 0.7397 |
| IX.I26–I28 | 0.7399 | *0.7474* | 0.6948 | 0.5785 | **0.7499** | 0.6977 |
| IX.I30–I5A | 0.8624 | 0.8624 | 0.7955 | 0.6321 | **0.8665** | 0.8010 |
| I35 | 0.7981 | **0.8050** | 0.7365 | 0.6246 | *0.7983* | 0.7423 |
| I350 | 0.8266 | **0.8346** | 0.7672 | 0.6112 | *0.8268* | 0.7784 |
| I359 | 0.8444 | *0.8525* | 0.7397 | 0.6020 | **0.8585** | 0.7490 |
| I42 | *0.8772* | **0.8838** | 0.8396 | 0.7193 | 0.8731 | 0.8458 |
| I420 | 0.8914 | *0.8986* | 0.8463 | 0.7816 | **0.9198** | 0.8590 |
| I428 | *0.8851* | **0.8921** | 0.8311 | 0.7144 | 0.8807 | 0.8392 |
| I429 | 0.8586 | *0.8611* | 0.8020 | 0.6408 | **0.8791** | 0.8116 |
| I44 | 0.9019 | *0.9050* | 0.8439 | 0.7591 | **0.9091** | 0.8467 |
| I440 | 0.8984 | *0.9102* | 0.7545 | 0.6638 | **0.9156** | 0.7610 |
| I441 | 0.8860 | *0.9021* | 0.8229 | 0.6309 | **0.9147** | 0.8338 |
| I442 | 0.9113 | *0.9151* | 0.8498 | 0.7564 | **0.9194** | 0.8551 |
| I447 | 0.9389 | **0.9417** | 0.9275 | 0.8633 | *0.9397* | 0.9305 |
| I45 | *0.8407* | **0.8493** | 0.7766 | 0.6646 | 0.8392 | 0.7828 |
| I46 | 0.8336 | *0.8481* | 0.7697 | 0.6562 | **0.8523** | 0.7809 |
| I47 | 0.8051 | *0.8023* | 0.7409 | 0.6480 | **0.8140** | 0.7461 |
| I48 | 0.8837 | *0.8861* | 0.7852 | 0.6952 | **0.8927** | 0.7933 |
| I480 | 0.8059 | *0.8135* | 0.7301 | 0.6735 | **0.8196** | 0.7290 |
| I481 | 0.8809 | *0.8872* | 0.8091 | 0.7141 | **0.8875** | 0.8152 |
| I482 | 0.9228 | *0.9206* | 0.8040 | 0.7169 | **0.9345** | 0.8216 |
| I50 | *0.8720* | **0.8747** | 0.8166 | 0.6169 | *0.8720* | 0.8235 |
| IX.I60–I69 | 0.6694 | *0.6723* | 0.6290 | 0.5473 | **0.6823** | 0.6348 |
| IX.I70–I79 | 0.7416 | **0.7441** | 0.6942 | 0.5956 | *0.7440* | 0.6974 |
| IX.I80–I89 | 0.6890 | *0.6949* | 0.6349 | 0.5731 | **0.6985** | 0.6324 |
| IX.I95–I99 | 0.6772 | *0.6782* | 0.6224 | 0.5519 | **0.6943** | 0.6275 |

Table 4: Extended ICD-10 code prediction performance by hierarchical group. Comparison of proposed LAMAE and LAMAE$_E$ under linear probing and fine-tuning of the corresponding pretrained backbone.

| ICD hierarchy | Linear Probing | | Fine-Tuning | |
|---|---|---|---|---|
| | LAMAE | LAMAE$_E$ | LAMAE | LAMAE$_E$ |
| **IX** | 0.8345 | 0.8340 | **0.8495** | *0.8486* |
| IX.I05–I09 | 0.8317 | 0.8278 | **0.8509** | *0.8452* |
| I07 | 0.8772 | 0.8802 | *0.8904* | **0.9039** |
| I071 | 0.8830 | 0.8740 | **0.9044** | *0.8763* |
| I078 | 0.8818 | 0.8977 | **0.9119** | 0.9099 |
| I08 | 0.8272 | 0.8235 | **0.8490** | 0.8420 |
| I080 | 0.8345 | 0.8318 | **0.8539** | 0.8425 |
| I081 | 0.8704 | 0.8695 | **0.8936** | 0.8730 |
| I083 | *0.8852* | 0.8573 | 0.8794 | **0.9163** |
| IX.I10–I1A | 0.7549 | 0.7570 | **0.7690** | *0.7678* |
| I11 | 0.8103 | 0.8124 | **0.8343** | 0.8272 |
| I13 | 0.8651 | 0.8704 | **0.8830** | 0.8798 |
| I130 | 0.8629 | 0.8687 | **0.8817** | 0.8812 |
| I132 | 0.9009 | *0.9096* | **0.9119** | 0.9022 |
| IX.I20–I25 | 0.7995 | 0.8046 | **0.8282** | 0.8250 |
| I20 | 0.7880 | 0.7963 | *0.8040* | **0.8183** |
| I200 | 0.8236 | 0.8269 | **0.8445** | 0.8440 |
| I209 | 0.7925 | 0.8075 | *0.8197* | **0.8217** |
| I21 | 0.8229 | 0.8402 | **0.8773** | 0.8600 |
| I210 | 0.9330 | **0.9632** | 0.9615 | *0.9631* |
| I211 | 0.9370 | 0.9588 | **0.9749** | *0.9681* |
| I213 | 0.8649 | 0.8996 | **0.9248** | *0.9043* |
| I214 | 0.8043 | 0.8117 | **0.8584** | *0.8429* |
| IX.I26–I28 | 0.7399 | 0.7474 | *0.7617* | **0.7682** |
| IX.I30–I5A | 0.8624 | 0.8624 | **0.8771** | *0.8759* |
| I35 | 0.7981 | 0.8050 | **0.8244** | *0.8180* |
| I350 | 0.8266 | 0.8346 | *0.8528* | **0.8644** |
| I359 | 0.8444 | 0.8525 | **0.8681** | *0.8657* |
| I42 | 0.8772 | *0.8838* | **0.8880** | 0.8805 |
| I420 | 0.8914 | 0.8986 | **0.9170** | *0.8935* |
| I428 | 0.8851 | 0.8921 | **0.8958** | *0.8932* |
| I429 | 0.8586 | 0.8611 | **0.8809** | *0.8752* |
| I44 | 0.9019 | *0.9050* | **0.9097** | *0.9053* |
| I440 | 0.8984 | **0.9102** | 0.8922 | *0.9044* |
| I441 | 0.8860 | *0.9021* | **0.9052** | 0.8800 |
| I442 | 0.9113 | *0.9151* | **0.9184** | 0.9095 |
| I447 | 0.9389 | *0.9417* | **0.9452** | 0.9410 |
| I45 | 0.8407 | *0.8493* | **0.8519** | 0.8442 |
| I46 | 0.8336 | 0.8481 | *0.8613* | **0.8634** |
| I47 | 0.8051 | 0.8023 | *0.8106* | **0.8123** |
| I48 | 0.8837 | 0.8861 | **0.9016** | *0.8993* |
| I480 | 0.8059 | 0.8135 | *0.8201* | **0.8254** |
| I481 | 0.8809 | *0.8872* | **0.8902** | 0.8828 |
| I482 | *0.9228* | 0.9206 | **0.9312** | 0.9285 |
| I50 | 0.8720 | 0.8747 | **0.8892** | *0.8857* |
| IX.I60–I69 | 0.6694 | 0.6723 | *0.6836* | **0.6841** |
| IX.I70–I79 | 0.7416 | 0.7441 | **0.7545** | *0.7456* |
| IX.I80–I89 | 0.6890 | 0.6949 | *0.7121* | **0.7156** |
| IX.I95–I99 | 0.6772 | 0.6782 | **0.6963** | *0.6939* |

