# OpenReview forum: "From Leads to Latents: Attention-Driven Masked Autoencoder for ECG Times Series"
_ICLR.cc/2026/Workshop/LMRL — Submitted to ICLR 2026 Workshop LMRL_

### Official Review · Reviewer_cV4p · 2026-02-11
**Review for LAMAE**

**Rating:** 4
**Confidence:** 2

**Review:**

## Summary
This manuscipt descibed a Masked AutoEncoder with Latent Attention and mask objective function for ECG data modeling. This methods outperformed baselines on different benchmarks.

## Strengths
1. This paper built a MAE with a latent attention design with cross-lead connection learning for ECG data, which could be a backbone model for ECG related tasks.
2. Authors provided comprehensive experiments and evaluations.

## Weaknesses
1. Authors added a LA block between encoder and decoder of the MAE. The LA seems a regular multihead attention and the objective function is also a common function for masked learning. But authors did not clarify the type of the per-lead encoder. Is it a transformer, LSTM, or other models? If it is a transformer, there should also be self-attention blocks. Why does this extra LA block improve the performance. Is this because of the design of LA or just increased model parameter scale? Especially, authors mentioned LA moedels cross-lead connection. I would appreciate, if it is possible to clarify why LA improved this modeling.
2. The proposed method did not always outperformed baselines. These cases that LAMAE did not work well can be a good outlier case study. While, there are some metrics just minor improvement. Test of significance could prove these improvement.

## Clarity
1. The figure 1 shows the mask is applied to ECG inputs, but the eq (1) shows it is applied to the inputs to LA module, which is the output of the encoder.
2. Line 086, $L = \\{l_I, I_{II}, ...\\}$ demonstrates several unexaplained terms. Explain in appendix what are $l_I, I_{II}, ...$ may help improve clarity.
3. Line 133, the $T_{vis} = |T_{vis}|$ may be confused. I suggest to use a different notation here.
4. The results also shows a model LLMAE_E, it is not clear what it is.

---

### Official Review · Reviewer_mBk1 · 2026-02-17

**Rating:** 2
**Confidence:** 3

**Review:**

This work investigates reconstruction-based pretraining using ECG time series. Specifically, the authors use cross-lead redundancy by using weight-sharing in the (single-lead) encoder and decoder. while transformer layers are used on the latent tokens. The results indicate that significant improvements are obtained in the small-data regime.

Pro:
- The paper is well written for the most part, including the description of the architecture.
- The method appears to be capable of improving small-data performance.

Con:
- As far as I can tell, nearly no models included in the results table are described. For example, I don't know what the difference between LAMAE and LAMAE_E is, and the baselines MMVM, ind_p, and ind_s are all not introduced.
- The contribution of the paper is not clear to me: is it not rather the standard that attention is operating on latent representations? Similarly, is it not common that the encoder/projection layer is re-used across tokens (even on extremely large datasets, e.g. ViTs)?

Overall, for a short workshop paper I find the lack of methodological novelty not the biggest issue. However,  this combined with the fact that results are mostly uninterpretable due to missing information (as well as a spelling mistake in the title; time series*) makes the paper unfortunately have little value to the community in its current state.

---

### Meta-Review · Area_Chair_ABdd · 2026-02-25

**Recommendation:** Reject
**Confidence:** 5

**Metareview:**

This work does not pass the bar for the LMRL workshop.

---

### Decision · Program_Chairs · 2026-03-02

**Decision:**

Reject

**Comment:**

Please see the meta-review.